# High-Density Lipoprotein Particles and Their Relationship to Posttransplantation Diabetes Mellitus in Renal Transplant Recipients

**DOI:** 10.3390/biom10030481

**Published:** 2020-03-21

**Authors:** Sara Sokooti, Tamas Szili-Torok, Jose L. Flores-Guerrero, Maryse C. J. Osté, António W. Gomes-Neto, Jenny E. Kootstra-Ros, Hiddo J.L. Heerspink, Margery A. Connelly, Stephan J. L. Bakker, Robin P. F. Dullaart

**Affiliations:** 1Department of Internal Medicine, University Medical Center Groningen, University of Groningen, 9713 GZ Groningen, The Netherlands; t.szili-torok@umcg.nl (T.S.-T.); j.l.flores.guerrero@umcg.nl (J.L.F.-G.); m.c.j.oste@umcg.nl (M.C.J.O.); a.w.gomes.neto@umcg.nl (A.W.G.-N.); s.j.l.bakker@umcg.nl (S.J.L.B.); dull.fam@12move.nl (R.P.F.D.); 2Department of Laboratory Medicine, University Medical Center Groningen, University of Groningen, 9713 GZ Groningen, The Netherlands; j.e.kootstra@umcg.nl; 3Department of Clinical Pharmacy and Pharmacology, University Medical Center Groningen, University of Groningen, 9713 GZ Groningen, The Netherlands; h.j.lambers.heerspink@umcg.nl; 4Laboratory Corporation of America® Holdings (LabCorp), Morrisville, NC 27560, USA; connem5@labcorp.com

**Keywords:** HDL cholesterol, HDL particles, HDL size, posttransplantation diabetes mellitus, renal transplant recipients

## Abstract

High concentrations of high-density lipoprotein (HDL) cholesterol are likely associated with a lower risk of posttransplantation diabetes mellitus (PTDM). However, HDL particles vary in size and density with yet unestablished associations with PTDM risk. The aim of our study was to determine the association between different HDL particles and development of PTDM in renal transplant recipients (RTRs). We included 351 stable outpatient adult RTRs without diabetes at baseline evaluation. HDL particle characteristics and size were measured by nuclear magnetic resonance (NMR) spectroscopy. During 5.2 (IQR, 4.1‒5.8) years of follow-up, 39 (11%) RTRs developed PTDM. In multivariable Cox regression analysis, levels of HDL cholesterol (hazard ratio [HR] 0.61, 95% confidence interval [CI] 0.40–0.94 per 1SD increase; *p* = 0.024) and of large HDL particles (HR 0.68, 95% CI 0.50–0.93 per log 1SD increase; *p* = 0.017), as well as larger HDL size (HR 0.58, 95% CI 0.36–0.93 per 1SD increase; *p* = 0.025) were inversely associated with PTDM development, independently of relevant covariates including, age, sex, body mass index, medication use, transplantation-specific parameters, blood pressure, triglycerides, and glucose. In conclusion, higher concentrations of HDL cholesterol and of large HDL particles and greater HDL size were associated with a lower risk of PTDM development in RTRs, independently of established risk factors for PTDM development.

## 1. Introduction

Posttransplantation diabetes mellitus (PTDM) is one of the major complications following renal transplantation. PTDM may be associated with adverse effects on both short- and long-term outcomes in renal transplant recipients (RTRs), including graft failure, cardiovascular disease, and patient survival [1,2,3,4]. The reported incidence of PTDM varies from 7% to 39% at one year after transplantation and from 10% to 30% at 3 years post-transplantation [5]. There are similarities between the pathogenesis of PTDM and type 2 diabetes mellitus (T2DM) [6]. Alternations in both insulin secretion and insulin resistance may be considered as risk factor for developing PTDM [7]. Interestingly, although insulin resistance is involved in the pathogenesis of low concentrations of high-density lipoprotein (HDL) cholesterol, HDL itself may stimulate insulin secretion and prolong β-cell survival [8]. Many studies have shown an inverse association between HDL cholesterol and worsening of insulin resistance, modulation of glucose metabolism, and progression to T2DM [9,10,11,12,13]. Notably, HDL particles vary in size, density, and function, causing various HDL particles to differ in their associations with insulin secretion, resistance, and incident diabetes [14,15]. HDL remodeling is altered in insulin-resistant conditions and T2DM, resulting in decreased HDL cholesterol coinciding with less large and more small HDL particles [16,17]. Thus, the distribution of HDL particles is altered from large HDL particles which are rich in cholesteryl esters to small HDL particles in T2DM [18]. Among HDL particles, measured by nuclear magnetic resonance (NMR) spectroscopy, lower levels of large HDL particles and smaller HDL particle size were associated with the development of insulin resistance and T2DM, often independently of typical diabetes risk factors [19,20,21,22,23].

Identifying RTRs who are more likely to develop PTDM may help to provide early interventions that may reduce mortality and morbidity outcomes in RTRs. Little is currently known about the association between HDL particle characteristics and the risk for development of PTDM in RTRs. Therefore, the aim of our study was to determine the association between different HDL particles with the risk of future PTDM in RTRs.

## 2. Materials and Methods

### 2.1. Design and Study Population

For this study, all RTRs (aged ≥18 years) with a ≥1 year post-transplantation period were eligible for participation. The data were collected between November 2008 and June 2011, from the Tranplantlines Food and Nutrition Study (NCT02811835), during outpatient clinic visit at the University Medical Center Groningen (UMCG), Groningen, the Netherlands, as described previously [24,25]. Written, informed consent was obtained from 707 (87%) of the 817 initially invited RTRs; only subjects providing written, informed consent were included in this study. For the present study, we excluded patients with missing data on HDL indices (n = 239) and patients with diabetes or a history of diabetes at baseline (n = 117), leaving 351 RTRs eligible patients for analysis. The study was conducted according to the guidelines laid down in the Declaration of Helsinki, and all procedures involving human subjects/patients were approved by the Institutional Review Board (METc 2008/186).

### 2.2. Data Collection

The baseline measurements were performed during a morning visit to the outpatient clinic as described in detail previously [26]. Information on medication and medical history was derived from patient records. Alcohol consumption and smoking behaviour information was obtained by using a questionnaire. Physical activity was assessed by using the valid Short QUestionnaire to ASsess Health-enhancing physical activity (SQUASH) score in time multiplied by intensity [27]. Body height and weight were measured, and body mass index (BMI) was calculated as weight (kilograms) divided by height squared (meters). Blood pressure and heart rate were measured with a semiautomatic device (Dinamap1846; Critikon, Tampa, FL) every minute for 15 min in a half-sitting position. The average of the last 3 measurements was calculated as a blood pressure value. All RTRs were instructed to collect a 24 h urine sample according to a strict protocol on the day before their visit to the outpatient clinic. Blood was drawn in the morning after the completion of the 24 h urine collection and after 8–12 h fasting.

### 2.3. Laboratory Measurements

Plasma glucose was measured using an enzymatic hexokinase assay. Glycated hemoglobin (HbA1c) was assayed using the turbidimetric inhibition immunoassay (Roche Integra). C-reactive protein (CRP) was measured using an immunoturbidimetric assay (all Roche Modular P Chemistry platform, Roche Diagnostics, Mannheim, Germany). Serum creatinine was measured by using an isotope dilution mass spectrometry (IDMS) traceable enzymatic method on a Roche P-Modulator automated analyzer (Roche Diagnostics, Basel, Switzerland), cystatin C was measured using a particle-enhanced immuno assay (Gentian, Moss, Norway), and renal function was assessed by using the combined creatinine cystatin C-based Chronic Kidney Disease Epidemiology Collaboration (CKD-EPI) formula in order to calculate the estimated glomerular filtration rate (eGFR) [28]. Urinary albumin concentration was determined by nephelometry (Dade Behring Diagnostic, Marburg, Germany). Frozen EDTA plasma samples collected at baseline were sent to LabCorp for testing. Lipoprotein parameters were measured by ^1^H-NMR spectroscopy using a Vantera^®^ NMR Clinical Analyzer (LabCorp, Raleigh, USA [29,30]). Triglycerides (TG, mg/dL), total cholesterol (TC, mg/dL), and HDL cholesterol (HDL-C, mg/dL) were quantified by using a Lipid Panel Assay NMR platform. To this end, a Lipid Panel Assay for quantifying TG, TC, and HDL-C was developed using Partial Least-Squares (PLS) regression models that were trained to several hundred or several thousand 400 MHz proton NMR spectra from serum specimens for which TG, TC, and HDL-C were chemically measured. Using a PLS regression routine, the spectral information in the combined methylene and methyl region was trained against the chemical measurements where the information was connected through latent variables. Cross-validation was performed to optimize the regression model. Once trained with a sufficient number of specimens, for any test specimen spectrum, the NMR spectral information was then converted into lipid concentrations using the regression coefficients for the regression model. Triglyceride-rich lipoproteins (TRL) (24–240 nm), low-density lipoproteins (LDL) (19–23 nm), HDL (7.4–13.0 nm), and subclasses of HDL particles (small, medium, and large) were quantified using the amplitudes of their spectroscopically distinct lipid methyl group NMR signals [31]. HDL size was calculated using the weighted averages derived from the sum of the diameters of each subclass multiplied by its relative mass percentage. Total TRL, LDL, and HDL particles were calculated by the sums of the concentrations of the respective subclasses. Estimated ranges of particle diameter for the subclasses were as follows: large HDL, 9.6–13 nm; medium HDL, 8.1–9.5 nm; and small HDL, 7.4–8.0 nm. All lipoprotein parameters were measured using an optimized version (LP4 algorithm) of NMR LipoProfile Test [32]. The estimated diameters of the HDL subspecies were as follows: H7P, 12.0 nm; H6P, 10.8 nm; H5P, 10.3 nm; H4P, 9.5 nm; H3P, 8.7 nm; H2P, 7.8 nm; and H1P, 7.4 nm.

### 2.4. PTDM

PTDM was defined on the basis of the American Diabetes Association criteria when at least one of the following criteria was met: (1) classic symptoms of diabetes (e.g., polyuria, polydipsia, unexplained weight loss) plus a nonfasting plasma glucose concentration ≥ 11.1 mmol/L (200 mg/dL); (2) fasting plasma glucose (FPG) ≥ 7.0 mmol/L (126 mg/dL); (3) use of antidiabetes medication; or (4) HbA1c ≥ 6.5% (48 mmol/mol) [33,34]. PTDM was recorded until 30 September 2015. RTRs were censored for PTDM at the time of graft failure (i.e., when they returned to dialysis or received another kidney transplantation) or death.

### 2.5. Statistical Analyses

All analyses were conducted with the use of the statistical packages IBM SPSS (version 24.0.1; SPSS, Chicago, IL, USA) and STATA/SE (version 14; StataCorp, College Station, TX, USA). A two-sided *p*-value less than 0.05 was considered statistically significant. Baseline RTR characteristics were compared to those with and without incident PTDM using *t*-tests and Wilcoxon tests for continuous values with normal distribution and skewed distribution, respectively. Data with normal distribution were expressed as mean ± SD, whereas data with skewed distribution were expressed as median (interquartile range [IQR]). Categorical data were expressed by their percentages and were compared by means of chi-squared tests. In prospective analyses, the Kaplan–Meier method was used to estimate PTDM rates in HDL cholesterol, total HDL particles, different subclasses of HDL particles, and HDL size ranked from the highest to the lowest value in tertiles, and Log-rank test was used to compare the estimated differences. Cox proportional hazard regression was used to calculate hazard ratios (HR) for incident PTDM for each predictor for both tertiles of and log-transformed variables which were not normally distributed, to find out the most significant association of HDL subclass or subspecies with PTDM development. The proportional hazards assumption was tested for each predictor along with covariates to see if it was violated. All models were adjusted for age, sex, and BMI. Subsequently, we performed additive adjustments in Cox regression analyses to avoid too many covariates included, based on the number of events. In additive multivariable models, we adjusted for smoking behaviour, alcohol use, and SQUASH score (model 2); lipid-lowering medication use, antihypertensive medication use, prednisolone dose, calcineurin inhibitor use, and proliferation inhibitor use (model 3); eGFR, urinary albumin excretion, cytomegalovirus (CMV) infection, and time since transplantation (model 4); HbA1c (model 5). Lastly, in model 6, we performed additional adjustment for BMI, systolic blood pressure (SBP), FPG, and triglycerides.

## 3. Results

### 3.1. Characteristics of RTRs at Baseline

Baseline characteristics and baseline plasma lipids and lipoproteins of 351 RTRs are shown in Table 1 and Table 2, respectively. RTRs who developed PTDM (n = 39) had a higher BMI, a larger waist circumference, higher systolic and diastolic blood pressure and HbA1c at baseline in comparison with RTRs who did not develop PTDM (n = 312). They used prednisolone and calcineurin inhibitor more frequently (Table 1). Plasma triglycerides and H2P were higher, whereas HDL cholesterol, large HDL particles, HDL size, H7P, and H6P were lower at baseline in subjects who developed PTDM (Table 2).

### 3.2. Association of HDL Cholesterol and HDL Particle Characteristics with Incident PTDM

In total, 39 RTRs (11%) developed PTDM during a median follow-up of 5.2 years (IQR 4.1–5.8 years). The median time between transplantation and study baseline for the RTRs who developed PTDM was 5.0 years (IQR 1.8–12.0 years). PTDM risk was first compared among the tertiles of HDL cholesterol, total HDL particles, HDL subclasses, and HDL size by Kaplan–Meier analysis (Figure 1). All of these HDL variables were categorized in tertiles from the highest to the lowest values. HDL cholesterol, large HDL particles, and HDL size showed statistically significant associations with PTDM *(p* = 0.019, *p* = 0.004, and *p* = 0.004, respectively). Total HDL, medium HDL, and small HDL particle concentrations were not associated with PTDM development in Kaplan–Meier analysis (*p* = 0.440, *p* = 0.347, and *p* = 0.110, respectively).

Subsequently, we performed Cox proportional hazard regression analyses for HDL cholesterol, large HDL particles, HDL size, with incident PTDM (Table 3). Higher HDL cholesterol was associated with lower risk of PTDM in crude analyses (HR, 0.53; 95% confidence interval [CI], 0.36–0.80 per 1SD mg/dL; *p* = 0.002). After adjustment for age, sex, and BMI (model 1) the association remained statistically significant (HR, 0.55; 95% CI, 0.36–0.83 per 1SD mg/dL; *p* = 0.005). Adjustment for additional variables including alcohol consumption, smoking status, and physical activity (model 2), use of lipid-lowering medication, anti-hypertensive medication, prednisolone dose, calcineurin inhibitors, and proliferation inhibitors (model 3), eGFR, albuminuria, CMV infection, and time after transplantation (model 4), and HbA1c (model 5) did not attenuate the association between HDL cholesterol and PTDM. After full adjustment for age, sex, BMI, SBP, FPG, and TG (model 6), the negative association remained statistically significant (HR, 0.61; 95% CI, 0.40–0.94 per 1SD mg/dL; *p* = 0.024). When analyzed per tertile, HDL cholesterol, was also inversely associated with PTDM development. In crude analysis, large HDL particles were associated with PTDM development (HR, 0.66; 95% CI, 0.51–0.84 per log 1SD; *p* = 0.001). This association persisted after adjusting for age, sex, BMI, and other covariates. In the fully adjusted model, we also found an inverse association between large HDL particles and incident PTDM (HR, 0.68; 95% CI, 0.50–0.93 per log 1SD; *p* = 0.017). When analyzed per tertile, a lower amount of large HDL particles was also associated with increased risk of PTDM (Table 3). In crude analyses, greater HDL size was inversely associated with PTDM development (HR, 0.47; 95% CI, 0.31–0.72 per 1SD; *p* = 0.001). This association remained after adjustment for other covariates in all other models and analyses according to tertiles of HDL size (Table 3). All together, the risk of developing PTDM was about threefold higher in the lowest vs. the highest tertile of HDL cholesterol, large HDL particles, and HDL size.

### 3.3. Confounding Influence of Other Lipoproteins on the Association of Large HDL Particles with Incident PTDM

We observed a negative correlation between the concentrations of large and medium HDL particles and that of small HDL particles (r = −0.24, *p* < 0.001 and r = −0.39, *p* < 0.001, respectively), and a positive correlation between the concentrations of medium and large HDL particles (r = 0.089, *p* = 0.047) (Appendix A). To determine potential confounders in the regression models, further analysis were performed with joint HDL subclasses in the same model (Table 4). In the multivariabe adjusted model, the association between large HDL particles and risk of PTDM remained significant when taking account of medium and small HDL particles (HR, 0.68; 95% CI, 0.50–0.93; *p* = 0.014).

Additionally, because of the correlation between large HDL and LDL as well as TRL particles (Appendix A), Cox regression analyses with HDL subclasses were jointly included in the model with further adjustment for LDL particles, TRL particles, and both (Table 4). The association between large HDL particles and risk of PTDM remained significant in all models.

### 3.4. Association of Large HDL Subspecies (H7P and H6P) with Incident PTDM

Finally, analyses were performed for the seven subspecies of HDL and incident PTDM during follow-up in 351 RTRs without diabetes at baseline (Appendix A). H1P through H5P showed no associations with the development of PTDM. Although H7P was associated inversely with PTDM development in the crude analyses, the association did not remain significant after adjustment for additional covariates (Appendix A). On the other hand, lower H6P was associated with an increased risk of developing PTDM not only in the crude model but also after further adjustment for additional covariates (Appendix A). We found a strong inverse association between H6P and incidence of PTDM in model 6 (HR 0.68, 95% CI 0.49–0.95 per log 1SD; *p* = 0.024). Additionally, there was an association between H6P tertiles and the risk of developing PTDM (about fivefold among the lowest vs. the highest tertile of H6P).

## 4. Discussion

We report for the first time on the associations of HDL size and various HDL subclasses and subspecies, determined with a novel NMR-based algorithm, with PTDM development in RTRs. In the current prospective study, we found that HDL cholesterol was inversely associated with PTDM. Furthermore, large HDL particles and larger HDL size were also inversely associated with PTDM. However, there was no association between other measures of other particles, total HDL, medium HDL, and small HDL particles, and the development of PTDM. Additionally, in further analyses among large HDL subspecies, we found that H6P was associated with the development of PTDM.

Many epidemiological studies have reported that HDL cholesterol is inversely associated with incident T2DM [14,15,21,22,35,36]. Consequently, HDL cholesterol is included in risk scores, drawn from the Framingham Offspring Study Diabetes Mellitus and the Diabetes Prediction Model risk scores [10,37], and is found to be useful for the prediction of PTDM in RTRs [38]. Baloore et al. found that HDL cholesterol was inversely associated with both first and recurrent hyperglycemia after renal transplantation [39]. HDL cholesterol modulates glucose metabolism by various mechanisms, such as countering the deleterious effects of oxidized LDL on insulin production and activation of AMP-activated protein kinase [40,41]. Additionally, intravenous reconstituted HDL reduced plasma glucose via increasing insulin and activating AMP-activated protein kinase [13]. These earlier findings consistently suggest that higher HDL levels may be protective against T2DM and PTDM.

Of further note, HDL includes heterogeneous particles which may differ in their function to promote the reverse cholesterol transport pathway and to exert anti-inflammatory, antioxidant, and antidiabetic effects [42,43,44,45,46]. Inverse associations between large HDL particles and incident T2DM have been reported previously [14,15,22]. The studies did not find any association between small HDL and development of type 2 diabetes. Our results are consistent with those findings, but to our knowledge, the association of HDL subclasses with PTDM have not been investigated previously in a renal transplant population. Of further note, Tabara et al. found that large HDL particles were inversely associated with insulin resistance, which could be a mechanism explaining the association between large HDL particles and incident T2D [15]. Although they found a positive association between small HDL particles and insulin resistance, small HDL particles were not associated with incident T2DM. An association between higher concentrations of small HDL particles and incident of T2DM was reported in a prospective study among 26,836 women [21]. Similarly, Festa et al. reported that a higher concentration of small HDL particles was independently associated with increased risk of diabetes during a five-year follow-up [20]. The pathophysiological relationship between insulin resistance and HDL particle characteristics, measured by NMR, was tested by the euglycemic clamp technique, documenting that small HDL particles associated positively and large HDL particles associated inversely with insulin resistance [19].

The current study also shows that HDL particle size was inversely associated with risk of PTDM in RTRs. This was consistently reported in two previous general population-based studies [21,22]. Mora et al. found that smaller HDL size was associated with a 4.5-fold higher risk of T2DM in women, independently of risk factors including HbA1c [21]. In line with this, the association between HDL size and PTDM development was independent of HbA1c among RTRs in our study. Gravey et al. found a strong relationship between increased insulin resistance and smaller HDL size measured by NMR, which could point to a causal role of HDL size and PTDM development among RTRs [19]. HDL size represented a biomarker of HDL metabolism associated with PTDM development in our study. However, this association is likely to be secondary to the association of large HDL particles and incident PTDM because of the collinearity between HDL size and large HDL particles. All together, the relation of HDL size and large HDL particles with incident PTDM in RTRs could conceivably be explained, at least in part, by increased insulin resistance or impaired insulin secretion [47]. However, we observed no association of the total HDL particle concentration with incident PTDM, in apparent contrast with the report by Mora et al. [21].

Although various HDL subclasses are interdependent, and their concentrations are affected by triglyceride-rich apolipoprotein B-containing lipoproteins, the association between large HDL particles and PTDM development remained significant when taking account of medium HDL, small HDL, total LDL, and total TRL particles. In comparison, weaker associations of large HDL particles and stronger association of small and medium HDL particles with coronary heart disease (CHD) were observed after adjustment for HDL subclasses and LDL particles in two previous studies [48,49]. This may reflect the fact that large HDL particles are a strong predictor of PTDM development; those correlations did not change the association between large HDL particles and PTDM development in RTRs in our study. Our study investigated the relationship between large HDL subspecies, H7P and H6P, with the incidence of PTDM in RTRs. We found that higher H6P levels were associated with lower risk of developing PTDM both as a continuous and as a categorical variable, whereas H7P was not associated with PTDM development. This suggests that there is variation between larger HDL subspecies, which can be distinguished by their size in the ability to predict PTDM in RTRs. In a recent study, individuals with obesity had lower values of both H7P and H6P in comparison with lean adolescents. However, among obese people, individuals with high insulin resistance had a lower amount of H6P compared to insulin-sensitive adults. Moreover, homeostatic model assessment for insulin resistance (HOMA-IR) was strongly correlated with fraction 22 which contains H6P as the major HDL subspecies [50].

Alterations in HDL functional properties and HDL metabolism may conceivably explain, at least in part, the association of large HDL particles and HDL size with incident PTDM, as currently observed. Besides a key role of the reverse cholesterol transport pathway in atherosclerosis development [51,52,53], HDL-mediated cellular cholesterol efflux (CEC) is considered to be relevant in maintaining β-cell function [54]. Interestingly, CEC from THP1 macrophages to apoB-depleted plasma predicts incident PTDM [55]. Using J774 macrophages, CEC to apoB-depleted plasma was found to be positively correlated with large- and medium-sized HDL, HDL size, as well as HDL particle concentration [56,57], while in another report, CEC was also correlated with HDL particle concentration though not with HDL particle size [51]. Of further importance, β-cell function is likely to be related to HDL anti-oxidative function as well [54], which is in part determined by paraoxonase-1 (PON-1), an anti-oxidative enzyme which predominantly resides on large HDL particles [46,58,59]. However, serum PON-1 activity did not predict incident T2DM in the general population [60].

HDL remodelling is altered in hyperglycemic circumstances, among other processes, due to concerted actions of lipases, cholesteryl ester transfer protein (CETP), and phospholipid transfer protein (PLTP) [16,61,62]. Increased plasma CETP activity consequent to diabetes-associated hypertriglyceridemia results in lower HDL cholesterol and smaller HDL particles [16,61]. Likewise, CETP gene variants that give rise to lower plasma CETP mass result in higher HDL cholesterol concentrations [63]. Interestingly, administration of CETP inhibitors could lower diabetes risk [64]. Of further relevance, genetic variation in PLTP, a lipid transfer protein that is able to convert HDL into larger and smaller HDL particles, affects HDL particle distribution [65], whereas we proposed earlier that higher plasma PLTP activity may predict increased diabetes risk [66]. Taken together, it is obvious that the extent to which specific processes involved in HDL remodelling are involved in PTDM development requires further study. Also, an effect of HDL particles and HDL function on insulin secretion could be considered as a mechanism to explain the protective effects of large HDL particles against T2D and PTDM [67].

Our study has strengths and limitations. A strength of this single-center study includes a relatively large prospective cohort of RTRs who completed the endpoint evaluation after a median follow-up of 5.2 years. Furthermore, RTRs with transient posttransplantation hyperglycemia were excluded from our study by including only RTRs with a functioning graft more than 1 year after transplantation. On the other hand, the median time after transplantation in the RTRs was 5 years, which makes it not possible to extrapolate the current results to the RTRs at an earlier stage after transplantation. Another limitation is that oral glucose tolerance tests were not carried out in this single-center study, which could have resulted in an underestimation of PTDM frequency. We also did not measure insulin at basline, precluding to demonstrate precise interactions between HDL particle characteristics, insulin restistance, and insulin secretion and PTDM development. Of note, however, we applied the American Diabetes Association Criteria to diagnose PTDM. Finally, the RTRs included were mostly of north European descent, limiting extrapolation to other ethnicities.

## 5. Conclusions

Higher levels of HDL cholesterol and of large HDL particles and greater HDL size, as determined by NMR, were associated with a lower risk of developing PTDM in RTRs. Our findings warrant replication in other RTR cohorts and in subjects of different ethnicities.

## Figures and Tables

**Figure 1 biomolecules-10-00481-f001:**
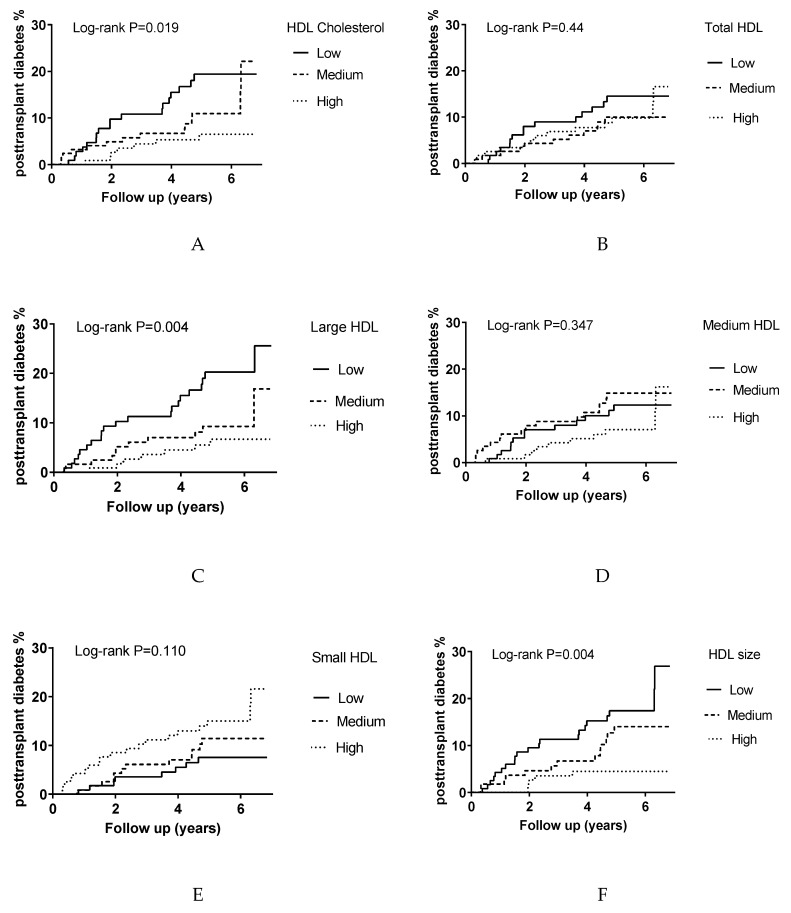
Kaplan–Meier curves for PTDM development according to the tertiles of HDL indices in 351 RTRs. Panel (**A**) for HDL cholesterol, panel (**B**) total HDL particles, Panel (**C**) for large HDL particles, Panel (**D**) for for medium HDL particles, Panel (**E**) for small HDL particles, and panel (**F**) for HDL size.

**Table 1 biomolecules-10-00481-t001:** Baseline Characteristics of 351 RTRs (Renal Transplant Recipients).

Variables	Total	Incident PTDM (Posttransplantation Diabetes Mellitus)		*p* Value
		Yes	No	
Participants, n	351	39	312	
General characteristics				
Men, %	54.1	61.5	53.2	0.325
Age, year (y)	51.3 ± 13.4	52.2 ± 10.4	51.2 ± 13.9	0.589
Lifestyle parameter				
Current smoker, %	14.0	17.9	13.4	0.445
Alcohol use, never, %	10.1	5.4	10.6	0.320
Physical activity score (time×intensity)	5640 (3220–8940)	4710 (2520–10440)	5760 (3305–8790)	0.597
Body composition				
Weight, kg	78.5 ± 16.1	85.5 ± 15.5	77.6 ± 15.9	0.005
Height, cm	173.8 ± 9.3	175.6 ± 9.4	173.6 ± 9.3	0.220
BMI, kg/m^2^	25.9 ± 4.5	27.7 ± 4.3	25.7 ± 4.5	0.010
Waist circumference, cm	96.2 ± 14.4	104.3 ± 14.1	95.3 ± 14.2	0.001
Transplant demographics				
Time since renal transplantation, y	4.9 (1.5–11.8)	3.2 (1.4–11.8)	5.0 (1.7–11.9)	0.297
Donor age, y	44.0 ± 15.2	44.4 ± 16.1	44.0 ± 15.2	0.888
Living donor, %	37.9	43.6	37.2	0.437
Dialysis duration, months	26(5–55)	21 (0–59)	26 (6–54)	0.604
Delayed graft function, %	8.0	12.8	7.4	0.237
Rejection, %	23.4	20.5	23.7	0.656
CMV infection, %	25.6	26.3	20.5	0.436
Blood pressure				
Systolic blood pressure, mmHg	135.8 ± 17.4	142.7 ± 15.9	135.0 ± 17.4	0.008
Distolic blood pressure, mmHg	82.8 ± 11.2	88.1 ± 11.6	82.1 ± 11.0	0.003
Hypertension, %	90.3	97.4	89.4	0.111
Glucose Homeostasis				
Glucose, mmol/L	5.1 ± 0.6	5.3 ± 0.6	5.1 ± 0.06	0.064
HbA1c, %	5.6 ± 0.3	6.0 ± 0.3	5.6 ± 0.3	< 0.001
Hs-CRP, mg/L	1.4 (0.6–3.8)	1.6 (0.8–2.8)	1.4 (0.6–4.2)	0.766
Renal function				
eGFR, mL/min per 1.73 m^2^	42.9 (30.4-56.8)	41.1 (25.0-52.3)	43.0 (31.0–57.8)	0.172
Urinary Albumin excretion, mg/24 h	38.9 (9.5–170.5)	43.2 (6.8–201.6)	38.5 (10.0–169.7)	0.615
Medication use				
Lipid-lowering medication, %	49.3	56.4	47.8	0.309
Anti-hypertensive medication, %	88.0	94.9	87.2	0.163
Prednisolone, mg/day	8.8 ± 1.8	9.3 ± 1.3	8.8 ± 1.9	0.024
Calcineurin inhibitor, %	58.1	76.9	55.8	0.012
Cyclosporine, %	41.9	53.8	40.4	
Tacrolimus, %	16.5	23.1	15.7	
Proliferation inhibitor, %	86.3	76.9	87.5	0.070
Azathioprine,%	19.9	15.4	20.5	
Mycophenolic acid, %	66.4	61.5	67.0	

Data are the mean ± SD, median (interquartile range) unless otherwise indicated. Significance was tested by *t*-tests and Wilcoxon tests for normal distribution and skewed distribution of continuous values respectively. BMI: body mass index; CMV: cytomegalovirus; HbA1c: glycated hemoglobin; Hs-CRP: high-sensitivity C-reactive protein; eGFR: estimated glomerular filtration rate.

**Table 2 biomolecules-10-00481-t002:** Baseline Lipids and Lipoproteins Values of 351 RTRs.

Variables	Total	Incident PTDM		*p* Value
		Yes	No	
Participants, n	351	39	312	
Triglycerides (total), mg/dL	149 (103–208)	173 (126–297)	146 (102–200)	0.004
Total cholesterol, mg/dL	198.9 ± 41.0	295.1 ± 45.8	198.1 ± 40.3	0.365
HDL cholesterol (total), mg/dL	54.4 ± 14.8	48.4 ± 10.6	55.2 ± 15.1	0.001
TRL particles (total), nmol/L	189 (138–265)	219 (148–316)	184 (136–263)	0.068
LDL particles (total), nmol/L	1400 (1171–1632)	1427 (1335–1673)	1395 (1158–1622)	0.212
HDL particles (total), µmol/L	20.5 ± 3.3	20.1 ± 3.2	20.5 ± 3.3	0.469
Large HDL particles, µmol/L	2.2 (1.3–3.6)	1.4 (1.0–2.3)	2.3 (1.4–3.8)	<0.001
Medium HDL particles, µmol/L	4.9 ± 2.1	4.6 ± 1.9	4.9 ± 2.2	0.452
Small HDL particles, µmol/L	13.0 ± 3.3	13.8 ± 2.9	12.9 ± 3.3	0.095
HDL size, nm	9.1 ± 0.5	8.9 ± 0.3	9.2 ± 0.5	0.001
HDL Subspecies				
H7P, µmol/L	0.3 (0.1–0.5)	0.2 (0.1–0.3)	0.3 (0.1–0.6)	0.022
H6P, µmol/L	0.6 (0.2–1.5)	0.3 (0.1–0.7)	0.7 (0.2–1.6)	0.002
H5P, µmol/L	0.9 (0.4–1.5)	0.8 (0.4–1.2)	0.9 (0.4–1.5)	0.378
H4P, µmol/L	1.7 (1.1–2.5)	1.7 (1.3–2.4)	1.7 (1.1–2.5)	0.841
H3P, µmol/L	2.8 (1.5–4.2)	2.9 (1.4–4.3)	2.8 (1.5–4.2)	0.768
H2P, µmol/L	9.6 (7.6–11.8)	11.0 (8.8–12.5)	9.4 (7.4–11.6)	0.024
H1P, µmol/L	3.0 (1.8–4.5)	3.0 (1.5–4.1)	3.0 (1.8–4.7)	0.569

Data are the mean ± SD, median (interquartile range) unless otherwise indicated. Significance was tested by *t*-tests and Wilcoxon tests for normal distribution and skewed distribution of continuous values respectively. HDL: high-density lipoprotein; TRL: triglyceride-rich lipoprotein; LDL: low-density lipoprotein.

**Table 3 biomolecules-10-00481-t003:** Association between HDL parameters and risk of PTDM in 351 RTRs.

Tertiles	1	2	3		
HDL cholesterol, mg/dL	>59	47–58	<47	Per 1SD	*p* value
Cases	7	14	18	39	
Crude analysis	1.00 (ref)	2.07 (0.84–5.14)	3.29 (1.37–7.88)	0.53 (0.36–0.80)	0.002
Model 1	1.00 (ref)	1.99 (0.79–5.05)	3.01 (1.22–7.43)	0.55 (0.36–0.83)	0.005
Model 2	1.00 (ref)	1.78 (0.69–4.63)	2.89 (1.16–7.23)	0.53 (0.34–0.83)	0.006
Model 3	1.00 (ref)	2.21 (0.85–5.74)	3.15 (1.26–7.92)	0.55 (0.36–0.83)	0.004
Model 4	1.00 (ref)	1.90 (0.74–4.90)	2.60 (1.02–6.61)	0.59 (0.39–0.91)	0.018
Model 5	1.00 (ref)	2.62 (1.01–6.80)	2.71 (1.05–6.99)	0.59 (0.38–0.92)	0.021
Model 6	1.00 (ref)	1.92 (0.76–4.90)	2.53 (1.00–6.48)	0.61 (0.40–0.94)	0.024
Large HDL particles µmol/L	>2.9	1.6–2.9	<1.6	Per 1SD Log	*p* value
Cases	7	11	21	39	
Crude analysis	1.00 (ref)	1.70 (0.66–4.39)	3.59 (1.53–8.46)	0.66 (0.51–0.84)	0.001
Model 1	1.00 (ref)	1.46 (0.55–3.85)	3.18 (1.29–7.87)	0.63 (0.47–0.84)	0.002
Model 2	1.00 (ref)	1.28 (0.47–3.47)	3.06 (1.22–7.66)	0.61 (0.44–0.84)	0.002
Model 3	1.00 (ref)	1.78 (0.66–4.80)	3.43 (1.38–8.52)	0.60 (0.45–0.81)	0.001
Model 4	1.00 (ref)	1.51 (0.55–4.10)	3.06 (1.18–7.88)	0.64 (0.47–0.86)	0.004
Model 5	1.00 (ref)	1.37 (0.51–3.73)	2.70 (1.05–6.91)	0.67 (0.48–0.93)	0.017
Model 6	1.00 (ref)	1.49 (0.53–3.94)	2.83 (1.10–7.29)	0.68 (0.50–0.93)	0.017
HDL size, nm	>9.2	8.9–9.2	<8.9	Per 1SD	*p* value
Cases	5	13	21	39	
Crude analysis	1.00 (ref)	3.05 (1.09–8.56)	4.57 (1.72–12.12)	0.47 (0.31–0.72)	0.001
Model 1	1.00 (ref)	2.78 (0.98–7.89)	4.09 (1.47–11.35)	0.48 (0.31–0.76)	0.002
Model 2	1.00 (ref)	2.60 (0.91–7.47)	3.68 (1.30–10.42)	0.50 (0.31–0.80)	0.004
Model 3	1.00 (ref)	3.56 (1.24–10.21)	4.63 (1.65–13.02)	0.48 (0.32–0.75)	0.001
Model 4	1.00 (ref)	2.90 (1.01–8.33)	3.80 (1.34–10.80)	0.51 (0.33–0.81)	0.004
Model 5	1.00 (ref)	2.10 (0.73–6.07)	3.01 (1.06–8.56)	0.62 (0.40–0.98)	0.040
Model 6	1.00 (ref)	2.85 (1.00–8.15)	3.46 (1.18–10.21)	0.58 (0.36–0.93)	0.025

HRs (95% CIs) were derived from Cox proportional hazard models. Multivariable model 1 was adjusted for age, sex, and BMI. Model 2 was adjusted for model 1 variables, alcohol consumption, smoking, and physical activity; Model 3 was adjusted for model 1 variables and treatment (lipid-lowering medication, anti-hypertensive medication, prednisolone dose, calcineurin inhibitors, and proliferation inhibitors); Model 4 was adjusted for model 1 variables and eGFR, urinary albumin excretion, CMV infection, time after transplantation; Model 5 was wadjusted for model 1 variables and HbA1c; Model 6 was adjusted for model 1 variables and systolic blood pressure, fasting plasma glucose, and triglycerides.

**Table 4 biomolecules-10-00481-t004:** Association Between Joint HDL Subclasses and Risk of Developing PTDM in 351 RTRs.

Joint HDL Subclasses	Large HDL Particles		Medium HDL Particles		Small HDL Particles	
Jointly Models *	HR (95% CI) Per Log1 SD	*p* Value	HR (95% CI) Per 1 SD	*p* Value	HR (95% CI) Per 1 SD	*p* Value
Undjusted	0.68 (0.50–0.93)	0.014	0.97 (0.68–1.38)	0.85	1.05 (0.51–2.15)	0.75
Adjusted for LDL particles	0.67 (0.49–0.92)	0.012	0.97 (0.68–1.39)	0.88	1.06 (0.77–1.47)	0.71
Adjusted for TRL particles	0.68 (0.50–0.93)	0.017	0.96 (0.67–1.37)	0.84	1.05 (0.75–1.46)	0.76
Adjusted for LDL and TRL particles	0.67 (0.49–0.92)	0.015	0.97 (0.68–1.38)	0.86	1.06 (0.76–1.47)	0.73

HRs (95% CIs) were derived from Cox proportional hazard models. * All models were adjusted for age, sex, BMI, systolic blood pressure, fasting blood glucose, and triglycerides, as well as for large, medium, and small HDL particles.

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
