# Peer review of "High-Density Lipoprotein Particles and Their Relationship to Posttransplantation Diabetes Mellitus in Renal Transplant Recipients"

_biomolecules, 2020, doi:10.3390/biom10030481_

Round 1

Reviewer 1 Report

The authors performed a very nice study using state of the art methodology. They provide a clear result, namely, that higher levels of HDL-C and large HDL-P as well as greater size of HDL particels are associated with a lower risk of PTDM development in RTRs. The paper is very well written, data clearly presented and perfectly discussed.

Reviewer 2 Report

The study performed by Sokooti et al. has investigated the association among the amount and size of lipoproteins and the risk of posttransplantation diabetes mellitus (PTDM). The authors provide evidence that the amount of HDL cholesterol and large HDLs are associated with a lower risk of PTDM in renal transplanted recipients (RTR). The main conclusion is supported by data and these findings deserve publication, but some important points should be addressed, particularly in HDL experiments.

It is not clear how the authors determined triglyceride, cholesterol and HDL cholesterol levels. NMR is rather used for determining lipoprotein size and relevant details of lipid calculations should be provided in methods.

The authors showed diabetic RTR data at baseline in Table 1 and 2, but they conducted the prospective investigation with non-diabetic patients (351) and it is not clear why they showed data from diabetic RTR at baseline.

The authors state that a limitation of the study is that oral glucose tolerance tests were not carried out. I found that insulin determinations would have been even more relevant since large HDLs may affect insulin secretion and this could be a main determinant of the associations that they observed. This point should be at least discussed.

HDL size and the amount of large HDLs are collinear variables and, for this reason, the association results are very similar. This point should be commented.

Reviewer 3 Report

Sokooti et al. show an excellent an interesting word about the HDL particles and their relationship to posttransplantation diabetes mellitus in renal transplant recipients. Redation and presentation of results is very well. However, to facility compresion of results, I suggest them to change the results of 3.2 section. I think that they should present a table to do easier results presented.
on the other hand, I suggest to delete table 5 because of lack of significant results (authors indicate in the section 33.5 that no significant inversely associations between H7P with PTDM).

Round 2

Reviewer 2 Report

No additional comments